# Comparison of gE/gI- and TK/gE/gI-Gene-Deleted Pseudorabies Virus Vaccines Mediated by CRISPR/Cas9 and Cre/Lox Systems

**DOI:** 10.3390/v12040369

**Published:** 2020-03-27

**Authors:** Jianglong Li, Kui Fang, Zhenxiang Rong, Xinxin Li, Xujiao Ren, Hui Ma, Huanchun Chen, Xiangmin Li, Ping Qian

**Affiliations:** 1State Key Laboratory of Agricultural Microbiology, Huazhong Agricultural University, Wuhan 430070, China; LJLHZAU0413@126.com (J.L.); fangkui1987@126.com (K.F.); rzx17806250660@163.com (Z.R.); lxxhzau@163.com (X.L.); renxujiao92@163.com (X.R.); qiagen@163.com (H.M.); chenhch@mail.hzau.edu.cn (H.C.); 2Laboratory of Animal Virology, College of Veterinary Medicine, Huazhong Agricultural University, Wuhan 430070, China; 3Key Laboratory of Preventive Veterinary Medicine in Hubei Province, The Cooperative Innovation Center for Sustainable Pig Production, Wuhan 430070, China

**Keywords:** CRISPR/Cas9, Cre/Lox, pseudorabies virus, vaccine

## Abstract

Pseudorabies (PR), caused by pseudorabies virus (PRV), is an acute and febrile infectious disease in swine. To eradicate PR, a more efficacious vaccine needs to be developed. Here, the gE/gI- and TK/gE/gI-gene-deleted recombinant PRV (rGXΔgE/gI and rGXΔTK/gE/gI) are constructed through CRISPR/Cas9 and Cre/Lox systems. We found that the rGXΔTK/gE/gI was safer than rGXΔgE/gI in mice. Additionally, the effects of rGXΔgE/gI and rGXΔTK/gE/gI were further evaluated in swine. The rGXΔgE/gI and rGXΔTK/gE/gI significantly increased numbers of IFN-γ-producing CD4+ and CD8+ T-cells in swine, whereas there was no difference between rGXΔgE/gI and rGXΔTK/gE/gI. Moreover, rGXΔgE/gI and rGXΔTK/gE/gI promoted a PRV-specific humoral immune response. The PRV-specific humoral immune response induced by rGXΔgE/gI was consistent with that caused by rGXΔTK/gE/gI. After the challenge, swine vaccinated with rGXΔgE/gI and rGXΔTK/gE/gI showed no clinical signs and viral shedding. However, histopathological detection revealed that rGXΔgE/gI, not rGXΔTK/gE/gI, caused pathological lesions in brain and lung tissues. In summary, these results demonstrate that the TK/gE/gI-gene-deleted recombinant PRV was safer compared with rGXΔgE/gI in swine. The data imply that the TK/gE/gI-gene-deleted recombinant PRV may be a more efficacious vaccine candidate for the prevention of PR.

## 1. Introduction

Pseudorabies (PR), also known as Aujeszky’s disease (AD), is an acute and febrile infectious disease in swine, and it is caused by pseudorabies virus (PRV) [1,2]. PR is characterized by reproductive failure, neurological disorders, and respiratory ailments in swine [3,4,5]. The disease can lead to substantial economic costs because of reproductive losses in sows and weight loss in PRV-infected adults in swine farmers [2]. Currently, some strategies based on the DIVA (differentiating infected from vaccinated individuals) vaccination program are applied to the eradication of PR in commercial swine populations [6]. A few developed countries, such as United States [3], several European countries [7], and New Zealand [8], have eradicated PR, whereas increased infection frequency and severity have again caused PR in swine and other animals since 2011 in China [9]. Therefore, a more efficacious vaccine is considered as a necessary tool to protect against PRV.

PRV is a member of the *Herpesviridae* family and consists of approximately 150 K double-strand DNA genome encoding ~70 proteins [3,10]. PRV genome contains a unique long region (UL) and a unique short region (US). The US region, which is bracketed by inverted repeat sequences, forms two possible PRV genome isomers opposite oriented US regions [3]. Genes, including protein kinase (PK), glycoprotein G (gG), gD, gI, and gE, are located within the US region [11]. Increasing evidence has shown that several proteins are closely related to the virulence of PRV, such as gE, gI, thymidine kinase (TK), and PK [12,13]. The gE is associated with cell fusion, virus diffusion between cells, neurotropism, and virion release. Moreover, gE, as a marker gene, is used to distinguish natural infection from vaccination when the vaccine lacks the gE gene [14]. TK is a viral enzyme to mediate PRV replication and spread in the central nervous system [15]. Up to date, all live PRV vaccine strains have been reported to contain one or more gene deletion [16,17]. Several vaccines based on PRV variants have been reported, such as gE/gI-gene-deleted PRV based on the Tianjin strain, inactivated gE/gI-gene-deleted PRV based on the ZJ01 strain, and TK/gE/gI-gene-deleted PRV based on the PR HN1201 strain [18,19,20]. Therefore, we further performed a comparison of gE/gI- and TK/gE/gI-gene-deleted PRV.

The clustered regularly interspaced short palindromic repeat (CRISPR)/Cas9 system is a robust and highly efficient tool for gene editing, which precisely manipulates specific genomic loci [21,22,23]. Previous studies have shown that gene editing in recombinants is found to be efficient and convenient within a short period by transfecting the CRISPR/Cas system [2,22,23,24]. Additionally, the Cre/lox system is a site-specific recombination system that also applied for gene manipulation [25]. A synaptic complex is formed by the association of two Cre-bound loxP sites [24,26,27]. The strand exchange is catalyzed through a mechanism shared by all of the tyrosine recombinases within the synaptic complex [28]. Therefore, it is interesting that gene-editing technology is used for vaccine development.

In the current study, the gE/gI- and TK/gE/gI-gene-deleted recombinant PRVs (rGXΔgE/gI and rGXΔTK/gE/gI) were developed using CRISPR/Cas9 and Cre/Lox systems. The effects of gE/gI- and TK/gE/gI-gene-deleted PRVs on protection against PRV were investigated in swine. The data indicate that the TK/gE/gI-gene-deleted PRV may be a more potentially effective vector to manufacture virus-vectored vaccines.

## 2. Materials and Methods

### 2.1. Viruses and Cells

The highly virulent PRV variant GX, which was adapted to and plaque-purified in PK-15 cells, was isolated from an affected pig farm in Guangxi, China, in 2016. HEK293T and PK-15 cells were cultured in Dulbecco’s modified Eagle’s medium (Invitrogen, Carlsbad, CA, USA) containing 10% heat-inactivated fetal bovine sera (FBS), streptomycin (100 μg/mL), and penicillin (100 IU/mL) at 37 °C in 5% CO_2_.

### 2.2. Plasmid Construction, PCR Amplification and Viral Genomic Preparation

The sgRNAs were designed according to the online CRISPR Design Tool (https://wwws.blueheronbio.com/external/tools/gRNASrc.jsp), and targeted the gE, gI, and TK gene open reading frames. The sequences of sgRNAs are listed in Table 1. The pX335 plasmid (Add gene, Beijing, China) was digested using the *BbsI* enzyme (New England Biolabs, Beijing, China), and CRISPR/Cas9 constructs were constructed. The primers (Table 1) were designed to amplify gE, gI, and TK homologous arms from the PRV GX strain using PCR, respectively. The GFP were inserted into the TK position, and mCherry were inserted into the gI and gE positions. The TKhm1-loxP-CMV-GFP-SV40polyA-loxP-TKhm2 and gIhm-loxN-CMV-mCherry-SV40polyA-loxN-gEhm donor templates were constructed by overlapping PCR. PRV genomic DNA was extracted through the TIANamp virus DNA kit (TIANGEN, Beijing China). The PCR primer synthesis and DNA sequencing in this study were performed by TsingKe Biotech Co. Ltd. (Beijing, China).

### 2.3. DNA Transfection and Purification of Recombinant Virus

The PRV GX genome was extracted as previously described [29,30]. To generate the recombinant PRV (rGXΔgE/gI or rGXΔTK/gE/gI), HEK293T cells were co-transfected with 1 µg of the PRV genome, 500 ng of cas9 plasmid sgRNA-gE and sgRNA-gI (or sgRNA-gE, sgRNA-gI, and sgRNA-TK), 200 ng of fragment gIhm-loxN-CMV-mCherry-SV40polyA-loxN-gEhm (or TKhm1-loxP-CMV-GFP-SV40polyA-loxP-TKhm2 and gIhm-loxN-CMV-mCherry-SV40polyA-loxN-gEhm) using Lipofectamine 2000 (Invitrogen, Carlsbad, CA, USA) for 48 h. Then, the cells were collected and subjected to three freeze–thaw cycles. Recombinant PRV was purified from the cell lysates using plaque purification in PK-15 cells overlaid with 1% low-melting-point agarose and 2% FBS in PBS. After 10 rounds of purification, the plaques were measured via fluorescent microscopy.

### 2.4. Cre-Mediated Recombination In Vitro

Conditions for Cre-mediated recombination in vitro were described in a previous study [31]. Briefly, 2 µg of rGXΔgE/gI or rGXΔTK/gE/gI genome, 50 mM Tris-HCI, pH 7.5/33 mM NaCI/10 mM MgCl2 and 8 units of Cre recombinase (New England Biolabs, Beijing, China) were mixed and incubated at 37 °C for 30 min. The reactions were stopped by heating the samples at 70 °C for 10 min to inactivate Cre, and then HEK293T cells were transfected with DNA. Fluorescent gene excision was developed by three rounds of plaque purification, as previously described [24].

### 2.5. In Vitro Growth Properties

One-step growth kinetics and plaque sizes of all the viruses were detected in this study, as previously described [20].

### 2.6. Animal Experiments

The 6-week-old female Balb/c mice (20–25 g) were purchased from the Laboratory Animal Research Center of Huazhong Agricultural University (Wuhan, China). The 5-week-old crossbred weaning piglets were purchased from the experimental farm of Huazhong Agricultural University. All experimental protocols were conducted according to the Research Ethics Committee of College of Veterinary Medicine, Huazhong Agricultural University, Hubei, China (No. 42000600035617, 17 September 2019).

Experiment 1: The BALB/c mice were randomly divided into 9 groups (*n* = 5/group). The mice in Groups A, B, C, or D were intraperitoneally inoculated with 100 μL of different doses (10^2^,10^3^, 10^4^, or 10^5^ TCID50) of rGXΔgE/gI. The mice in Groups E, F, G, or H were intraperitoneally inoculated with 100 μL of different doses (10^2^,10^3^, 10^4^, or 10^5^ TCID50) of rGXΔTK/gE/gI. Mice in Group I were injected with PBS serving as control. Then, clinical signs were monitored daily. At 14 days post-inoculation, all surviving mice were euthanatized, and the brain and lung tissues were collected.

Experiment 2: Pigs were randomly divided into three groups (*n* = 4/group). The piglets were seronegative for PRV, which were identified using a commercially available PRV-gB antibody detection kit (Combined Biotech Co., Ltd., Shenzhen, China). The pigs in Group A or B were intramuscularly immunized with 10^6^ TCID50 rGXΔgE/gI or rGXΔTK/gE/gI. Pigs in Group C were treated with 1 mL of PBS serving as control. The pigs were boosted with the same dose at 21 days post-immunization (dpi) and intranasally challenged with 10^7^ TCID50 virulent PRV GX strain at three weeks after the booster vaccination. Clinical signs were recorded daily for up to 16 days.

### 2.7. Flow Cytometry

The frequencies of IFN-γ-producing CD3^+^CD4^+^ and CD3^+^CD8^+^ T-cells from the CD3^+^ lymphocytes in the blood were analyzed using flow cytometry. Samples were collected at 14 days after the booster immunization. Then, sample processing and flow cytometry analysis were carried out as described in a previous report [32]. Mononuclear cells from the blood were isolated based on the previous description [33]. The mononuclear cells were stimulated in vitro with inactivated PRV (MOI = 1) for 17 h before intracellular staining, as previously described [32].

### 2.8. PRV-gD Specific Antibodies Measurement 

The levels of PRV-gD specific antibodies in the serum were measured using an indirect enzyme-linked immunosorbent assay (ELISA). Briefly, the 96-well flat-bottomed microtiter plates were incubated with 0.5 μg per well of purified gD protein in coating buffer (pH 9.5) at 4 °C overnight, and blocked with 1% bovine serum albumin at 37 °C for 1 h. Then, the serially diluted serum samples were added into the plate and incubated for 1 h at 37 °C. Horseradish peroxidase (HRP)-conjugated goat anti-swine IgG (1:5000) (AntGene Bio Co., Ltd.,Wuhan, China) was used to cover the plate for 1 h at 37 °C. The serum PRV-gD specific antibody titers were measured at wavelength 450 nm. The antibody endpoint titer was calculated based on the highest dilution, which gave an OD450 twice that of the naïve group without dilution. 

### 2.9. Serum Neutralisation Test

Serum neutralizing antibody titer was detected as previously described [34,35]. Briefly, the serum samples were diluted using PBS. The 96-well flat-bottomed tissue culture plates (Thermofisher, Waltham, MA, USA) were covered with the diluted serum samples and supplemented with a viral suspension with a titer of 200 TCID50 PRV GX strain in 50 μL. After incubation for 1 h at 37 °C, the plates were incubated with 50 μL of the PK-15 cell suspension for 3 days. Finally, the PRV-specific neutralizing antibody titer was analyzed and expressed as the reciprocal of the highest dilution while PK-15 cell infection was inhibited.

### 2.10. Virus Isolation

Rectal and nasal swabs were collected everyday post-immunization or post-challenge, and then the virus was isolated according to the previous description [18].

### 2.11. Hematoxylin and Eosin (HE) Staining

Brain and lung tissues were fixed in 10% neutral-buffered formalin, subjected to paraffin embedding, and then cut into 4-μm thick slices. The slices were deparaffinized, rehydrated and stained with hematoxylin and eosin. The photograph was taken under a light microscope.

### 2.12. Statistical Analysis

All data were presented as the mean ± standard deviation (SD) and analyzed using Graphpad Prism 6.0 software. Comparisons were conducted via one-way ANOVA, followed by Tukey’s test. A *p* value less than 0.05 was considered as statistically significant.

## 3. Results

### 3.1. Generation and Identification of gE/gI (or TK/gE/gI)-Deleted Recombinant PRVs via a CRISPR/Cas9- and Cre-lox-Based System

In order to obtain gE/gI (or TK/gE/gI)-deleted recombinant PRV (rGXΔgE/gI or rGXΔTK/gE/gI), CRISPR/Cas9 and Cre-lox-based systems were applied into producing new PRV virulent variants. The amino acid sequences of major antigens, gB and gD of PRV GX, are shown in Figure 1A,B, and they exhibited some variations and deletion compared with the Bartha strain and previous pandemic Ea strain by sequence alignment. 

We then established an express vaccine development strategy using two highly efficient gene edit systems, the CRISPR/Cas9 and Cre/Lox systems. HEK293T cells were co-transfected with the PRV genome, cas9 plasmid sgRNA-gE and sgRNA-gI (or sgRNA-gE, sgRNA-gI and sgRNA-TK), fragment gIhm-loxN-CMV-mCherry-SV40polyA-loxN-gEhm (or TKhm1-loxP-CMV-GFP-SV40polyA-loxP-TKhm2 and gIhm-loxN-CMV-mCherry-SV40polyA-loxN-gEhm). Then, the cells were collected and subjected to single-cell FACS technique to purify recombinant PRV. Further, the Cre/Lox system was used to facilitate fluorescent marker genes excision. GFP (or GFP and mCherry) genes were flanked with LoxP and Lox N pairs, respectively (Figure 1C, Figure 2A). Fluorescence detection revealed that the recombinant viruses (rGXΔgE/gI or rGXΔTK/gE/gI) expressing red or/and green fluorescence were successfully visualized (Figure 1D, Figure 2B). Subsequently, the recombinant PRV (rGXΔgE/gI or rGXΔTK/gE/gI) were collected and infected HEK293T cells. HEK293T cells with fluorescence were analyzed using the single-cell FACS technique and plated one cell per well to a 96-well plate pre-cultured with HEK293T cells. Fluorescence detection was conducted (Figure 1E, Figure 2C). Then, one round of plaque purification was performed to obtain the pure recombinant viruses when the wells displayed maximum fluorescence overlapping signals (Figure 1F, Figure 2D). In addition, PCR amplification confirmed the purity of the recombinant virus. The results demonstrated that recombinant PRV (rGXΔgE/gI) showed TK amplification, whereas gE/gI amplification was negative (Figure 1H). Similarily, TK and gE/gI gene amplification exhibited negative in recombinant PRV (rGXΔTK/gE/gI; Figure 2F). Finally, the fluorescence markers were removed in vaccine candidates due to vaccine safety concerns and regulation. Cre-treated recombinant viruses were obtained and infected to HEK293T cells. The fluorescent gene was excised through four rounds of plaque purification (Figure 1G, Figure 2E). Further, one-step growth kinetics demonstrated that the rGXΔgE/gI and rGXΔTK/gE/gI strains propagated slightly slower than the parental strain GX (Figure 2G). These results indicated that the gE/gI (or TK/gE/gI)-deleted recombinant virus was successfully generated using the CRISPR/Cas9- and Cre-lox-based systems.

### 3.2. Safety of rGXΔgE/gI and rGXΔTK/gE/gI in Mice

We then evaluated the safety of rGXΔgE/gI and rGXΔTK/gE/gI in mice. Results revealed that morbidity and mortality of mice immunized with rGXΔTK/gE/gI or PBS were not affected. However, the rGXΔgE/gI-inoculated mice displayed high morbidity and mortality (Table 2). Additionally, HE staining demonstrated that no histopathological lesions were observed in mice immunized with rGXΔTK/gE/gI or PBS. In contrast, purkinje neuron injury in the brains, and slight hemorrhages and congestion in the lungs were observed and obvious in mice immunized with the rGXΔgE/gI group (Figure 3A,B). These findings suggest that rGXΔTK/gE/gI is safer than rGXΔgE/gI in mice.

### 3.3. The Production of IFN-γ-Producing CD4^+^ and CD8^+^ T-Cells

To examine the amount of IFN-γ-producing CD4^+^ and CD8^+^ T-cells, which are critical for virus elimination, flow cytometry analysis was performed. We found that rGXΔTK/gE/gI and rGXΔgE/gI significantly elevated the IFN-γ-producing CD4^+^ and CD8^+^ T-cells in pigs. However, the numbers of IFN-γ-producing CD4^+^ and CD8^+^ T-cells were not affected between the rGXΔTK/gE/gI group and the rGXΔgE/gI group. (Figure 4A,B). The data imply that rGXΔTK/gE/gI and rGXΔgE/gI immunizations induce equivalent PRV-specific T-cell immune responses.

### 3.4. Immunogenicity of rGXΔgE/gI and rGXΔTK/gE/gI in Pigs

In order to monitor PRV gD-specific antibody responses and NAbs against the PRV GX strains, the serum samples were collected at 0, 14, 28, 42, 56, and 63 after vaccination. ELISA analysis demonstrated that there was no difference shown in PRV gD-specific antibody from rGXΔTK/gE/gI and rGXΔgE/gI groups at 0, 14, 28, 42, 56, and 63 days post-immunization (Figure 5A). Moreover, neutralizing antibody assay revealed that the levels of NAbs against the GX strain were similar in the two vaccinated groups (Figure 5B). These results indicate that rGXΔTK/gE/gI and rGXΔgE/gI immunizations induce equivalent PRV-specific humoral immune responses.

### 3.5. Protection of Pigs Immunized with rGXΔgE/gI and rGXΔTK/gE/gI from Virulent Challenge

In order to evaluate the protection efficacy of the two genetically deleted PRV strains against lethal GX challenge, pigs were challenged intranasally with a highly virulent PRV GX strain. No clinical signs were observed in pigs immunized with rGXΔgE/gI and rGXΔTK/gE/gI after virulent challenge. However, pigs in the PBS group displayed typical PR signs (depression, anorexia, cough, diarrhea, and systemic neurological signs) with high fever (40.5–42 °C) from 2 dpc till death (Figure 6A). Meanwhile, the fever frequencies were the highest in the PBS group (21/25; Table 3). In addition, all pigs died within 10 dpc in the PBS group, while pigs were survived in other groups (Figure 6B). The challenge virus was isolated from the nasal and rectal swabs of unvaccinated pigs at 1–16 dpc. In contrast, no viral shedding was detected in pigs vaccinated with rGXΔTK/gE/gI and rGXΔgE/gI (Table 3).

Further, after challenge, the HE staining revealed that the pigs in the PBS group suffered severe microscopic pathological lesions in multiple organs, such as obvious non-suppurative meningoencephalitis and hemorrhages in the brains, and congestion in the lungs. Moreover, the pigs in the rGXΔgE/gI-vaccinated group still had slight histopathological lesions, including meningoencephalitis, hemorrhages, and congestion in the lungs. In contrast, the pigs immunized with rGXΔTK/gE/gI did not show pathological lesions (Figure 7A,B).

## 4. Discussion

In the current study, we constructed the gE/gI- and TK/gE/gI-gene-deleted recombinant PRVs (rGXΔgE/gI and rGXΔTK/gE/gI) using CRISPR/Cas9 and Cre/Lox systems. The comparison of rGXΔTK/gE/gI and rGXΔgE/gI mediated by CRISPR/Cas9 and Cre/Lox systems was evaluated. rGXΔTK/gE/gI was demonstrated to be safer than rGXΔgE/gI in mice. The rGXΔTK/gE/gI and rGXΔgE/gI significantly promoted PRV-specific T-cell immune response and humoral immune response in swine. Further analysis revealed that rGXΔTK/gE/gI was safer compared with rGXΔgE/gI in swine.

Although there were significant efforts to control and eliminate PR, the disease has been endemic in swine in many places [36]. More effective PRV vaccines need to be explored to control the disease. Interestingly, it has been reported that piglets immunized with rPRVTJ-delgE were protected, whereas incomplete protection was provided by the Bartha-K61 vaccine [18], suggesting that rPRVTJ-delgE can update Bartha-K61 for the control of the currently epidemic PR. Notably, in our study, we isolated the PRV-GX strain and developed a gene-deleted vaccine for this variant. Several steps of single gene recombination and marker gene excision were performed to delete multiple genes to develop PRV vaccines. A PRV double gene deletion vaccine candidate was obtained, which involved approximately twenty rounds of time-consuming plaque purification [37]. In this study, to promote the multi-gene editing efficiency in viral genomes, two highly efficient gene edit systems, the CRISPR/Cas9 system and the Cre/Lox system, were combined. Further single-cell FACS technology was used to elevate virus purification efficiency. The gE/gI/TK and gE/gI genes were deleted, and marker genes from recombinant viruses rGXΔTK/gE/gI (GFP and mCherry) and rGXΔgE/gI (mCherry) were also excised using the Cre/lox site-specific recombination system. The procedure was simple and easy to carry out, requiring only the readily harvested Cre protein [38]. Subsequently, the gE/gI- and gE/gI/TK-gene-deleted PRVs (rGXΔgE/gI or rGXΔTK/gE/gI) were developed via the CRISPR/Cas9 and Cre/Lox systems.

Previously, gE-deleted PR vaccines were found to be safe and efficacious for the control and eradication of PR [39]. The TK gene, as one of the first PRV genes, was responsible for virulence [40]. Accumulating evidence has shown that TK, gI, and gE genes are considered as major virulence determinants of PRV genome [41,42,43,44]. It has been reported that deleting the TK gene decreases virus replication in the nervous system and the ability to cause encephalitis [45,46]. In addition, the deletion of TK and gE/gI genes caused the reduction of virulence and attenuation of PRV, but did not affect the immunogenicity of PRV [18,47,48]. Interestingly, in this study, we demonstrate that the LD50 of rGXΔTK/gE/gI was higher than 10^5^ TCID50 in mice, whereas the LD50 of rGXΔgE/gI in mice was (10^3.68^ TCID50) less than 10^5^ TCID50, suggesting that rGXΔTK/gE/gI may be safer than rGXΔgE/gI in mice. Notably, Wang et al. discovered that viral shedding was not showed in all pigs vaccinated with rPRVTJ-ΔgE [18]. Consistently, this study verified that no other clinical signs associated with PRV infection and viral shedding were observed in pigs vaccinated with rGXΔTK/gE/gI and rGXΔgE/gI after challenge. However, after challenge, the pigs in rGXΔgE/gI-vaccinated group had slight histopathological lesions compared with the pigs immunized with rGXΔTK/gE/gI. We speculate that virulence of rGXΔgE/gI may be stronger than that of rGXΔTK/gE/gI, leading to the histopathological lesions in the rGXΔgE/gI-immunized group. It is important that a gene-deleted PRV can retain immunogenicity following the gene deletion [47]. In the current study, we demonstrate that the gE/gI- and gE/gI/TK-gene-deletions do not impair the immunogenicity of the virus in pigs. The PRV-specific cellular immune response and humoral immune response induced by rGXΔTK/gE/gI were consistent with those caused by rGXΔgE/gI in pigs. Taken together, these results imply that rGXΔTK/gE/gI may be a better candidate for the protection of PRV.

## 5. Conclusions

In conclusion, we combined CRISPR/Cas9 system and Cre/Lox system to generate gE/gI- and gE/gI/TK-gene-deleted variants. The pathogenicity and immunogenicity were evaluated in susceptible animals. The findings indicate that the recombinant virus protects pigs against PR, and rGXΔTK/gE/gI may be a promising vaccine vector and better control of prevalent PR.

## Figures and Tables

**Figure 1 viruses-12-00369-f001:**
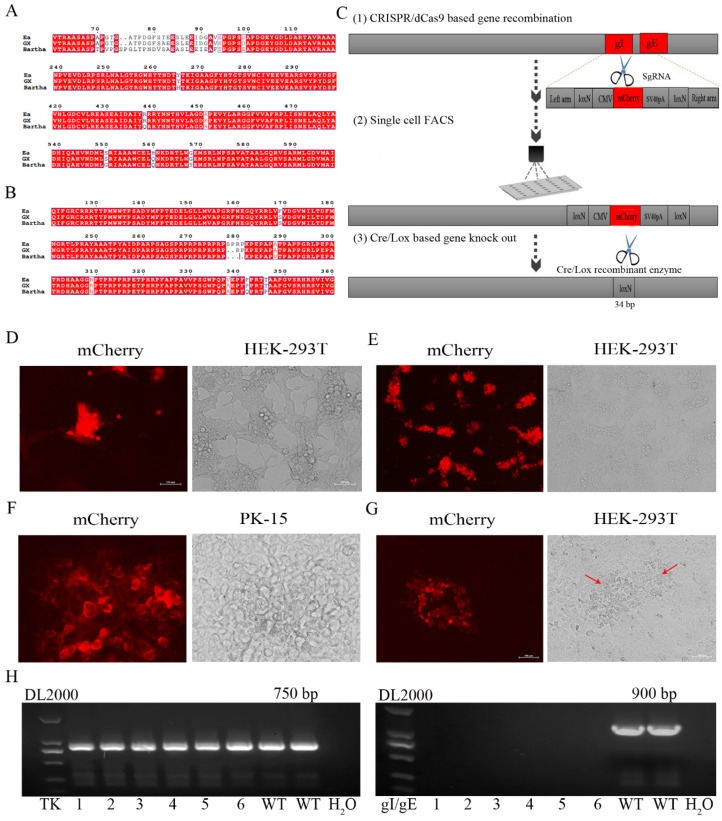
The gE/gI-gene-deleted recombinant PRV (rGXΔgE/gI) was developed using CRISPR/Cas9 and Cre/Lox systems. (**A**,**B**) Amino acid sequence alignments of PRV gB and gD (PRV GX strain, PRV Bartha strain, and the previous pandemic strain PRV Ea). (**C**) The vaccine development strategy was performed. PRV virulent gene gE and gI were replaced with mCherry via CRISPR/Cas9 system-assisted homologous recombination. Recombinant virus purification was accelerated using single-cell FACS. The mCherry gene was deleted through the Cre/Lox system. (**D**) Fluorescence detection was performed when HEK293T cells were co-transfected with pX335-Cas9-sgRNA-gE, pX335-Cas9-sgRNA-gI, gIhm-loxN-CMV-mCherry-SV40polyA-loxN-gEhm donor template and PRV genome. (**E**) Single cells with fluorescent gene recombinant PRV were selected using FACS. (**F**) Plaque purification of gE/gI-gene-deleted recombinant PRV in agarose-DMEM plates. (**G**) Plaque purification for fluorescent gene excision PRV following Cre- recombinase treatment. Arrow indicated cell plaques. (**H**) PCR verification of gE/gI gene deletion.

**Figure 2 viruses-12-00369-f002:**
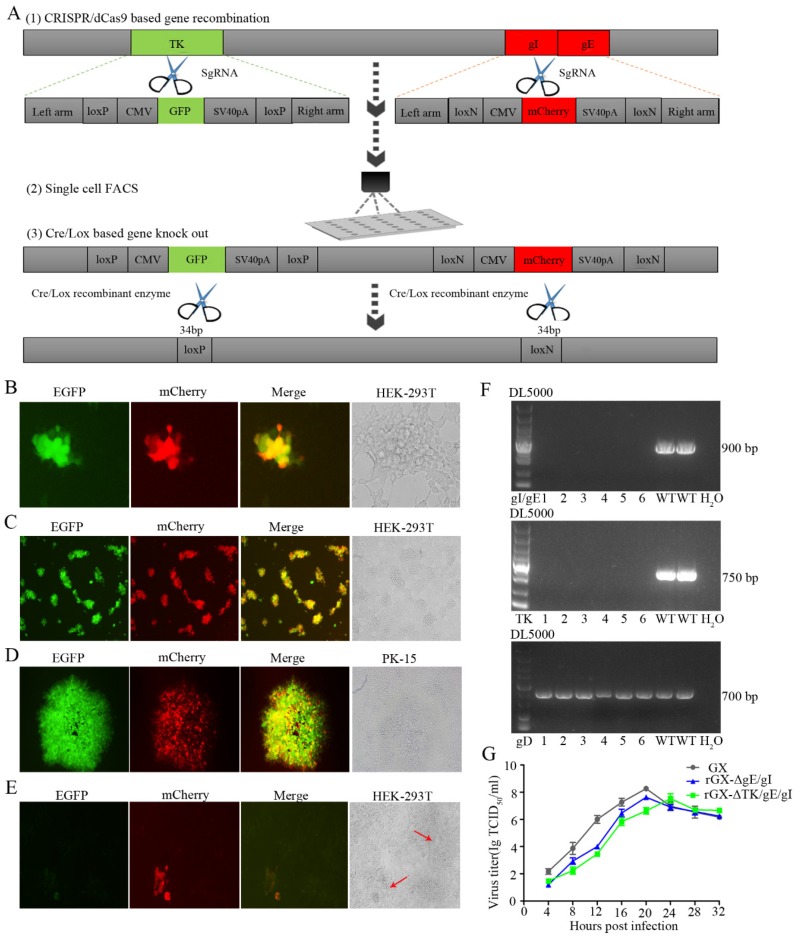
The gE/gI/TK-gene-deleted recombinant PRV (rGXΔTK/gE/gI) was developed using CRISPR/Cas9 and Cre/Lox systems. (**A**) Overview of the strategy for vaccine development. PRV virulent gene TK and gE/gI were replaced with GFP and mCherry respectively via CRISPR/Cas9 system-assisted homologous recombination. Recombinant virus purification was accelerated using single-cell FACS. The GFP and mCherry gene were deleted through the Cre/Lox system. (**B**) Fluorescence detection was conducted while HEK293T cells were co-transfected with pX335-Cas9-sgRNA-TK, pX335-Cas9-sgRNA-gE, pX335-Cas9-sgRNA-gI, TKhm1-loxP-CMV-GFP-SV40polyA-loxP-TKhm2, and gIhm-loxN-CMV-mCherry-SV40polyA-loxN-gEhm donor templates, PRV genome. Scale bar, 100 μm. (**C**) Single cells with fluorescent gene recombinant PRV were sorted through FACS. Scale bar, 100 μm. (**D**) Plaque purification of gE/gI/TK-gene-deleted recombinant PRV in agarose-DMEM plates. Scale bar, 100 μm. (**E**) Plaque purification for fluorescent genes excision PRV following Cre–recombinase treatment. Arrows indicate cell plaques. Scale bar, 100 μm. (**F**) PCR verification of TK and gE/gI gene deletion. (**G**) One-step growth curves of the recombinant viruses in BHK-21 cells.

**Figure 3 viruses-12-00369-f003:**
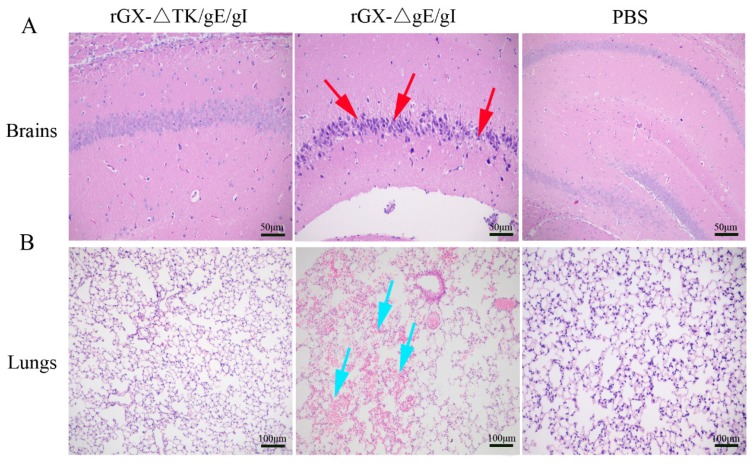
The effect of the gE/gI- and gE/gI/TK-gene-deleted PRVs on histopathological lesions in mice. (**A**,**B**) Pathological lesions of brain and lung tissues were detected using HE staining. Scale bar, 50 or 100 μm. Arrows indicate the cell infiltrates. HE, hematoxylin and eosin staining.

**Figure 4 viruses-12-00369-f004:**
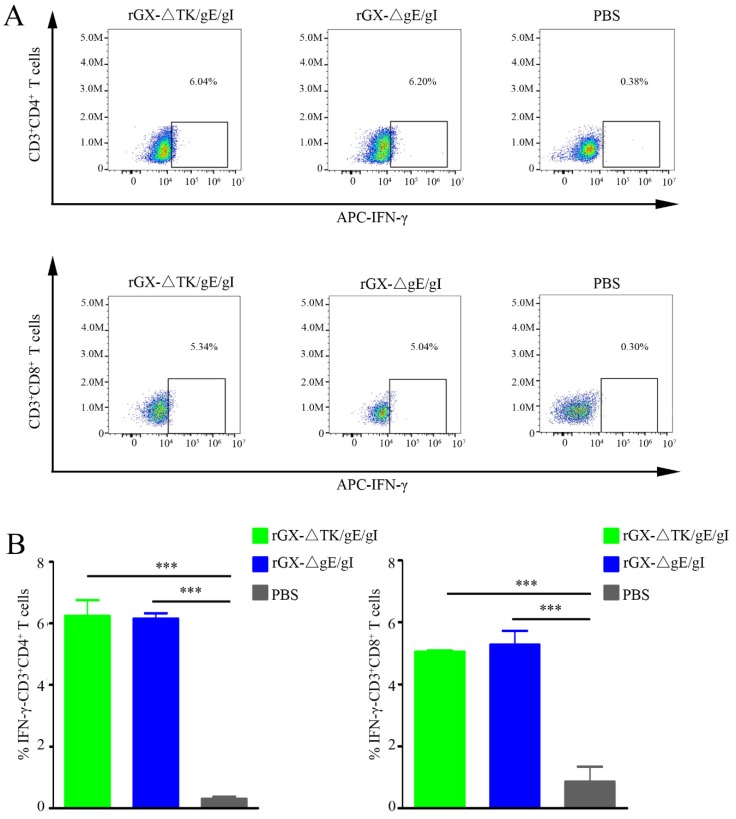
The rGXΔgE/gI or rGXΔTK/gE/gI facilitated production of IFN-γ-producing CD4^+^ and CD8^+^ T-cells in the blood. (**A**) Flow cytometry analysis was used to measure the IFN-γ-producing CD4^+^ and CD8^+^ T-cells. (**B**) The calculation of IFN-γ-producing CD4^+^ and CD8^+^ T-cells. *n* = 4. *** *p* < 0.001.

**Figure 5 viruses-12-00369-f005:**
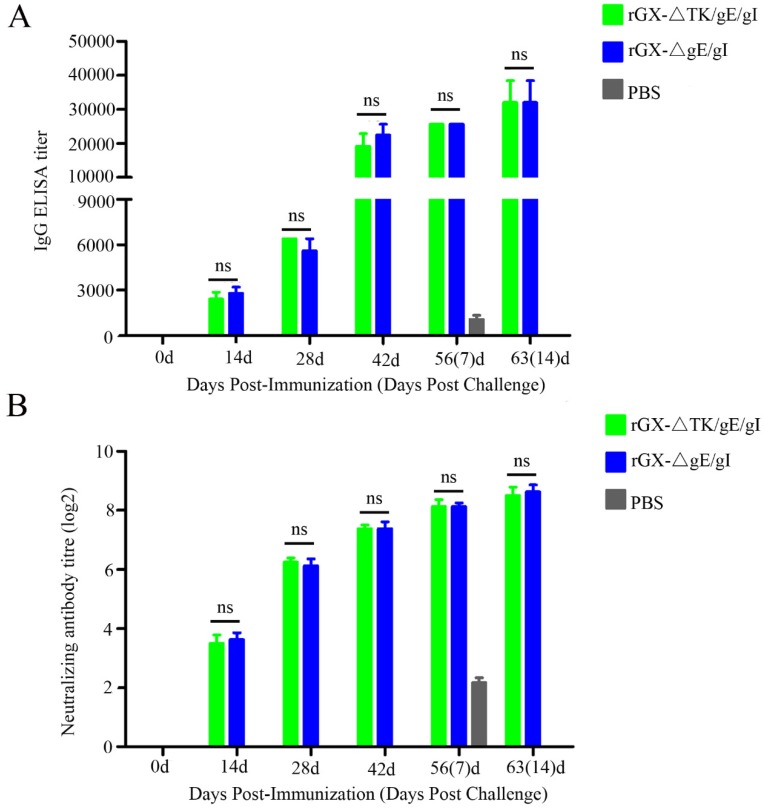
PRV-gD-specific IgG titre and neutralizing antibody titre were evaluated. (**A**) The indirect ELISA was used to detect gD-specific IgG titer from serum samples at 0, 14, 28, 42, 56, and 63 dpi. (**B**) Neutralizing antibody titer was examined in the serum of the immunized pigs. The neutralizing ability of antisera generated against PRV-GX strain was calculated and presented as the log2 of the reciprocal of the highest serum dilution when PK-15 cell infection was inhibited. *n* = 4. ns represents not significant.

**Figure 6 viruses-12-00369-f006:**
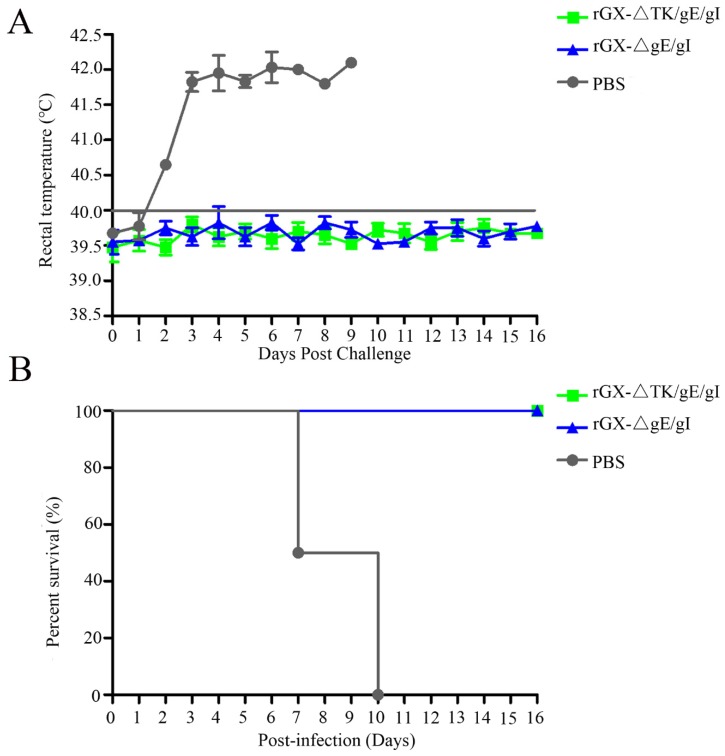
The effect of rGXΔgE/gI and rGXΔTK/gE/gI on rectal temperature and survival rates in pigs after challenge. Pigs were challenged with virulent PRV GX strain (10^7^ TCID50) at 49 dpi, then (**A**) rectal temperature was recorded and (**B**) survival rates were calculated. Rectal temperature ≥40.0 °C was fever. *n* = 4.

**Figure 7 viruses-12-00369-f007:**
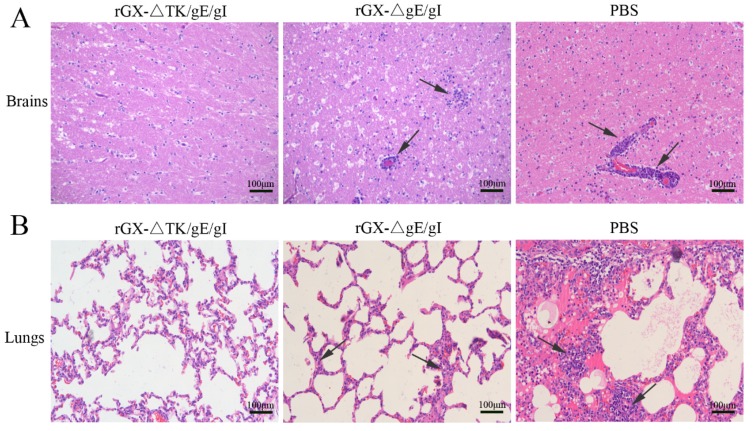
The effect of the gE/gI- and gE/gI/TK-gene-deleted PRV on histopathology in the brain and lung tissues of pigs. (**A**,**B**) HE staining was carried out to measure pathological lesions of brain and lung tissues in pigs. Scale bar, 50 or 100 μm. Arrows indicate the cell infiltrates. HE, hematoxylin and eosin staining.

**Table 1 viruses-12-00369-t001:** Sequences that were used in this study.

Name	Sequences (5′-3′)	Application
TK HR L arm	F:AAGCTTCGCCGTGGTCGTCACGCCCATGAAGGTGCG R:TTAATTAAGGCGCCGTCGAGGTAGATCCGGAGGATGCG	TK hm1amplification
TK HR R arm	F:TTAATTAATGCGCCTTCACGTCGGAGATGGGGGTGTGAC R:AAGCTTGGTGCCGTTGGGGAAGAGCATCAGGGCCTTG	TK hm2amplification
gI HR L arm	F:AAGCTTGTGCCCGCGCCGACCTTCCCCCCGCCCGCG R:TTAATTAAGCGGTGGCGCGAGACGCCCGGCGCG	gI hm1amplification
gE HR R arm	F:TTAATTAAGTCCCGCCCCGCTTAAATACCGGGAG R:AAGCTTACGTCCAGGGCGTCGGCGTCCGTCAGCCCG	gE hm1amplification
GFP	F:ATCGATATGGTGAGCAAGGGCGAGGAGCTG R:AGATCTTCACTTGTACAGCTCGTCCATGCCG	GFP amplification
mCherry	F:ATCGATATGGTGAGCAAGGGCGAGGAGGATAAC R:AGATCTTCACTTGTACAGCTCGTCCATGCCG	mCherry amplification
sgRNA-TK	F:CACCGCTGGCGCGCTTCATCGTCGGGG R:AAACCCCCGACGATGAAGCGCGCCAGC	TK sgRNA cloning
sgRNA-gE	F:CACCGAACGCCACCGCGGACGAGTCGG R:AAACCCGACTCGTCCGCGGTGGCGTTC	gE sgRNA cloning
sgRNA-gI	F:CACCGGCGTACTCGCGCGTGTAGCAGG R:AAACCCTGCTACACGCGCGAGTACGCC	gI sgRNA cloning
gI/gE	F:GCCGACGACCCCCGCGCCCCCCCGGGGGAC R:ACGTCCAGATCCCGGCCAGCACGGCGCCGTC	gI/gE gene verification
TK	F:CCGGTATTTACGATGCGCAGACCCGGAAGC R:CCTCCATGCCGCGCGCCTGCGCCGCCACGG	TK gene verification
gD	F:TGCCCGCGCCGACCTTCCCCCCGCCCGCGTAC R:GCGTACGGCGTGGCGGCGGCGTAGGCCCGCGG	gD gene verification

**Table 2 viruses-12-00369-t002:** Outcome of infection with rGX△TK/gE/gI and rGX△gE/gI in mice.

Groups	Doses (TCID50)	Amounts	Morbidity	Mortality	LD50 (TCID50)
rGX-△TK/gE/gI	10^5^ 10^4^ 10^3^ 10^2^	5 5 5 5	0/5 0/5 0/5 0/5	0/5 0/5 0/5 0/5	
rGX-△gE/gI	10^5^ 10^4^ 10^3^ 10^2^	5 5 5 5	5/5 4/5 2/5 0/5	5/5 3/5 1/5 0/5	10^3.68^
PBS	0.1 ml	5	0/5	0/5	

**Table 3 viruses-12-00369-t003:** Disease outcome in pigs inoculated following challenge with the highly virulent strain PRV-GX.

Group	Fever (≥40.5 °C)	Days to Fever Onset	Fever Frequency	Survival Rate	Viral Shedding
PBS	4/4	2	21/25 ^a^	0/4	4/4
rGX-△gE/gI	0/4	-	0/64	4/4	0/4
rGX-△TK/gE/gI	0/4	-	0/64	4/4	0/4

Groups of pigs (*n* = 4) were inoculated with 10^6^ TCID50 of rGX-△TK/gE/gI or rGX-△gE/gI or PBS and then challenged with 1 mL 10^7^ TCID50/100 μL of the PRV-GX strain at 49 days post-inoculation (dpi). Following challenge, fever, days to fever onset, fever frequency, and survival were recorded. Fever is defined as rectal temperature ≥40.5 °C. ^a^, Days with fever/total days observed.

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
