# Peer review of "Comparison of gE/gI- and TK/gE/gI-Gene-Deleted Pseudorabies Virus Vaccines Mediated by CRISPR/Cas9 and Cre/Lox Systems"

_viruses, 2020, doi:10.3390/v12040369_

Round 1

Reviewer 1 Report

The manuscript provides strong evidence that deleting the gene encoding thymidine kinase (TK) from a prototypic PRV vaccine design improves the safety of the vaccine without hampering its efficacy, at least within the context of a virus deleted for the genes encoding gE and gI. I have no major concerns regarding the data, but significant concerns regarding presentation. Manuscript requires additional proofing for proper grammar. I noted some specific corrections for grammar below, but I was not comprehensive in these edits. A thorough proofing is essential before manuscript should be considered for publication.

The study should also be placed into context with current vaccines and their complications. In particular, the Discussion would be improved by including an explanation as to why the Bartha vaccine is not safe enough despite having an intact TK gene. Have Bartha-vaccinated pigs been found to suffer from vaccine-related illness?

Additional major points:

1) I do not understand why gB and gD sequences are provided in figure 1, given that purpose of this figure is to document gE and gI deletions. How are gB, gD, gE, and gI related in this regard?

2) Figure 1 legend requires additional information. In panel D, there is mention of sgRNAs but no mention of how is Cas9 supplied. In panel G, there is no mention of how Cre is supplied. In panel H what is the difference between the left and right panels, and what is “DL200”?

3) Description of the method (lines 175-178) is quite unclear and concepts are jumbled. More description is needed, but equally important a rationale flow of the description is required.

4) What does “selection markers” refer to (line 189)? This requires explanation.

5) Need explanation of why gE/gI is deleted again in figure 2. In other words, why wasn’t TK gene deleted from the virus made in figure 1?

6) Figure 2 legend requires additional information. In panel F, what is DL5000? In panel G, what cell type is used?

7) Figures 1 and 2: Absence of PCR products is insufficient for validations. Sequence confirmation and Western blots are required. gE antibodies are readily available. If TK antibodies are not available, confirmation can instead be acquired by demonstrating that TK deletion virus fails to grow in non-dividing cells.

8) Figure 6B: if each data set consists of four pigs each, then why does the PBS control only have 2 deaths on the plot?

9) Figure 3 & 7: Legends indicate blue stain represents damaged tissue, but I think it would be more appropriate to say the blue stain represent cell infiltrates.

10) Lines 59-67: The message of this paragraph is that the authors think their method is interesting. That is a rather inconsequential point that would be better off moved to the discussion or removed altogether.

A few examples of grammar and spelling that require proofing (not comprehensive):

Line 51: Change “neurotropic tropism” to “neurotropism”

Line 68: Change “In current study” to “In the current study”

Line 69: Change “through” to “using a”

Line 84: “fhomologous” (typo)

Line 172: Change “were showed” to “are shown”

Line 175: “efcient” (typo)

Reviewer 2 Report

This study generated a new vaccine candidate for the PRV GX isolate by deleting viral proteins TK, gE, and gI. The authors tested the deletion mutants in mice and pigs to demonstrate immunogenicity and protection.

However, this study lacks novelty for the following reasons:

  1. The original PRV vaccine, PRV Bartha, from 1961 contains a spontaneous deletion in gE and gI. Thus, gE/gI deletion is the archetype for most PRV vaccine candidates. Vaccine candidates deleting TK, and TK in combination with gE (and gI) have been developed since the 1980s-90s. There have been numerous publications in the last 5 years generating vaccine candidates from PRV field isolates in China by deleting combinations of gE/gI and TK (Hu et al., 2015, Yang et al., 2016, Wang et al., 2014, Cong et al., 2016, Tong et al., 2016, Zhang et al., 2015, Wu et al., 2016). Thus, the authors claim that "comparison of double and triple gene-deleted PRV is still not available" is not accurate, and the introduction does not provide sufficient background and does not include all relevant references.
  2. The approach using CRISPR/Cas, single cell FACS, and Cre-Lox marker excision is identical to the approach published in Liang et al Scientific Reports 2016, limiting the novelty of its use here. The authors do not cite Liang et al in the introduction or explain that this approach has been developed and published previously, which seems a serious omission.
  3. The author's principal conclusions, that these vaccine candidates are immunogenic, protective, and that adding the TK deletion improves safety in mice is already well-established in the literature.

Consequently, I have scored Originality/Novelty, Significance, Interest, and Overall Merit very low.

The overall quality of writing is acceptable, but there are several instances of misspellings and repeated words (e.g. lines 175, 300, 309).

Reviewer 3 Report

The present manuscript shows a study aimed to the comparison of two gene-deleted pseudorabies virus vaccine through CRISPR/Cas9. Seems that the authors completely followed the recommended article structure.

The research is well conducted, and the results are very interesting and useful, after other investigation.

However, the manuscript should be revised in the language, that is very poor.
